https://doi.org/10.1038/s42003-023-05662-9　　OPEN
# Susceptibility to acute cognitive dysfunction in aged mice is underpinned by reduced white matter integrity and microgliosis

Dáire Healy[1], Carol Murray[1], Ciara McAdams[1], Ruth Power [1], Pierre-Louis Hollier[1], Jessica Lambe[1], Lucas Tortorelli[1], Ana Belen Lopez-Rodriguez[1] & Colm Cunningham [1✉]

Age is a significant but heterogeneous risk factor for acute neuropsychiatric disturbances such as delirium. Neuroinflammation increases with aging but the determinants of underlying risk for acute dysfunction upon systemic inflammation are not clear. We hypothesised that, with advancing age, mice would become progressively more vulnerable to acute cognitive dysfunction and that neuroinflammation and neuronal integrity might predict heterogeneity in such vulnerability. Here we show region-dependent differential expression of microglial transcripts, but a ubiquitously observed primed signature: chronic *Clec7a* expression and exaggerated *Il1b* responses to systemic bacterial LPS. Cognitive frailty (vulnerability to acute disruption under acute stressors LPS and double stranded RNA; poly I:C) was increased in aged animals but showed heterogeneity and was significantly correlated with reduced myelin density, synaptic loss and severity of white matter microgliosis. The data indicate that white matter disruption and neuroinflammation may be key substrates of the progressive but heterogeneous risk for delirium in aged individuals.

[1] School of Biochemistry & Immunology, Trinity Biomedical Sciences Institute & Trinity College Institute of Neuroscience, Trinity College Dublin, 152-160Pearse St. Dublin 2, Dublin, Republic of Ireland. ✉email: colm.cunningham@tcd.ie

Ageing is a highly heterogeneous, progressive and deleterious loss of function resulting from the life-long accumulation of molecular and cellular defects[1,2], which vary significantly among individuals of a given chronological age, influenced by genetic and lifestyle factors[3]. The decline in functional capacity contributes to increased vulnerability to morbidity and this age-dependent physiological deterioration also includes negative impacts on the brain, including significant reductions in size, composition, vasculature and plasticity[4,5]. This not only leads to cognitive impairment per se but also increases vulnerability to acute neuropsychiatric disruptions of function, such as delirium, upon physiological perturbations such as infection, injury and surgery. However, the prevalence and severity of such cognitive vulnerability has been shown to vary greatly between individuals of a given chronological age. Some individuals, despite advancing chronological age, show successful, resilient, ageing while others of the same age will age poorly, falling into a state of increased risk of falls, physical disability and cognitive impairment. This decreased resilience to acute stressors is known as frailty[3]. Indeed, 38% of 60–78 year olds, with no distinct pathology, have demonstrated a significant decline in cognitive performance[6]; specifically within episodic memory processing, working memory, spatial memory, processing speed and implicit memory function[7]. Frailty status is associated with a significantly greater vulnerability to cognitive dysfunction and delirium under an acute stressor[8–14].

Inflammation is now widely recognised as a significant risk factor for dementia and a driver of the ageing process[15,16]. Inflammaging, the chronic low-grade inflammation associated with the ageing process[17], may contribute to neuroinflammation. Microglia are the brain's resident immune cell population and are responsible for maintaining the homeostasis of the CNS environment. However, under chronic activation, in ageing and neurodegenerative conditions, these cells become primed. This microglial priming is characterised by changes in the activation state of the microglial cell in response to their chronic activation which renders it highly responsive to secondary systemic inflammatory challenge. In the ageing brain, such 'trained' responses may produce deleterious effects and contribute to pathology[18]. However, recent studies show heterogeneity between different brain regions in the responsiveness of microglia in both young and aging mice[19]. Differences between white matter and grey matter structures of the brain have also been identified with ageing resulting in greater microglial responses in white matter compared to grey[20] characterised by white matter-associated microglia (WAMs). Despite several differences, these microglia do show an overlapping transcriptional signature with so-called disease-associated microglia (DAMs) observed in multiple CNS degenerative pathologies in mice[21]. Those findings suggest that secondary inflammatory challenge may produce different inflammatory changes and cognitive or neurological sequalae in different areas of the brain.

Previous studies have demonstrated that in addition to advancing age[22], prior neurodegenerative pathology, synaptic loss[23,24] and loss of cholinergic innervation[25] underpin a heightened vulnerability of mice to acute cognitive dysfunction upon systemic inflammation induced by bacterial endotoxin (lipopolysaccharide; LPS). Furthermore, cumulative evidence derived from fMRI, PET imaging and psychopharmacological investigations has re-iterated the importance of loss of structural and functional integrity of neuroanatomical connections and neurotransmitter inadequacy in the development of delirium and dementia[26–29]. Nonetheless, our understanding of inflammation-induced delirium remains relatively poor and animal models offer opportunities to elucidate structural and cellular substrates of delirium risk. We have previously shown

that synaptic loss and neuroinflammatory activation significantly increase risk for delirium-like changes in animal models superimposing acute systemic inflammation upon underlying, evolving, neurodegeneration[23]. However, underlying risks for acute delirium-like events arising during ageing per se remain little investigated.

Here we characterise how markers of microglial activation and neuroinflammation change with age, in a region-dependent manner, and examine the hypothesis that these microglia show exaggerated innate immune responses to systemic challenge with LPS that may also be regionally heterogeneous. Furthermore, we hypothesised that advancing age would render animals increasingly vulnerable to acute cognitive dysfunction under systemic inflammatory conditions. We demonstrate that degree to which aged animals are susceptible to cognitive impairment correlates with the extent of their microgliosis, white matter integrity and pre-synaptic terminal density.

## Results

**Ageing confers increased vulnerability to exaggerated sickness behaviour**. Open field analysis of 25 and 8-month-old animals (female and male 8 months: $26 \pm 4$, $33 \pm 4$ g respectively; female and male 25 months: $30 \pm 3$, $34 \pm 1$ g respectively), treated with saline or LPS, revealed significant differences in sickness behaviour, at 3 hours post-challenge, as a function of age. The number of rears and squares crossed following treatments, as a percentage of their own baseline performance, assessed 24 hours before treatment, revealed that age conferred significant vulnerability to LPS-induced suppression of spontaneous activity as assessed by Bonferroni post-hoc analysis ($p < 0.05$) after significant main effects of age and treatment (Fig. 1a, b). Aged animals treated with LPS scored significantly lower than adult animals treated with LPS. That is, there were effects of age and treatment on the number of rears ($p < 0.001$; $F_{1,22} = 50.95$; $p < 0.01$, $F = 9.073$ respectively) and the distance covered ($p < 0.01$; $F_{1,22} = 11.98$, $p < 0.01$; $F_{1,22} = 8.778$ respectively) and post-hoc tests showed that LPS had larger effects in old than in young (Fig. 1a, b). Core body temperature was also measured 4 hours post-challenge with aged animals showing a significantly larger decrease from baseline in old animals than in young (interaction of age and treatment: $p < 0.05$ $F_{1,21} = 7.316$) (Fig. 1c). Since previous studies have reported that sickness behaviour may have different temporal profiles between young and aged animals that vary across recovery time[30] we analysed a separate cohort of mice over 24 h (supplementary fig. 1). These data show that the exaggerated effect of LPS in aged animals largely persists across 24 h.

Despite clear, statistically significant sickness behaviour responses to LPS only being evident in the aged LPS-treated population equivalent increases in circulating IL-1β protein levels were observed in both adult and aged LPS-treated animals (Fig. 1d). However, aged animals showed a greater increase in expression of both *Il1b* and *Il1a* RNA transcript levels compared to young-LPS-treated animals (Fig. 1e; interaction of age and treatment $p < 0.05$ $F_{1,21} = 4.186$, $p < 0.01$ F $_{1,22} = 8.054$ for *Il1b* and *Il1* respectively). These data demonstrate that 25-month-old animals are significantly more vulnerable to show heightened hippocampal IL-1 responses and acute illness responses to acute systemic inflammatory challenge.

**Regional differences in innate immune response**. We then assessed whether there were regional differences in microglial and inflammatory mediator responses to LPS during aging in these same animals (Fig. 2). Microglial markers were chosen based on their reported involvement in removal of neuronal debris and association with microglial phenotypes occurring in tissue

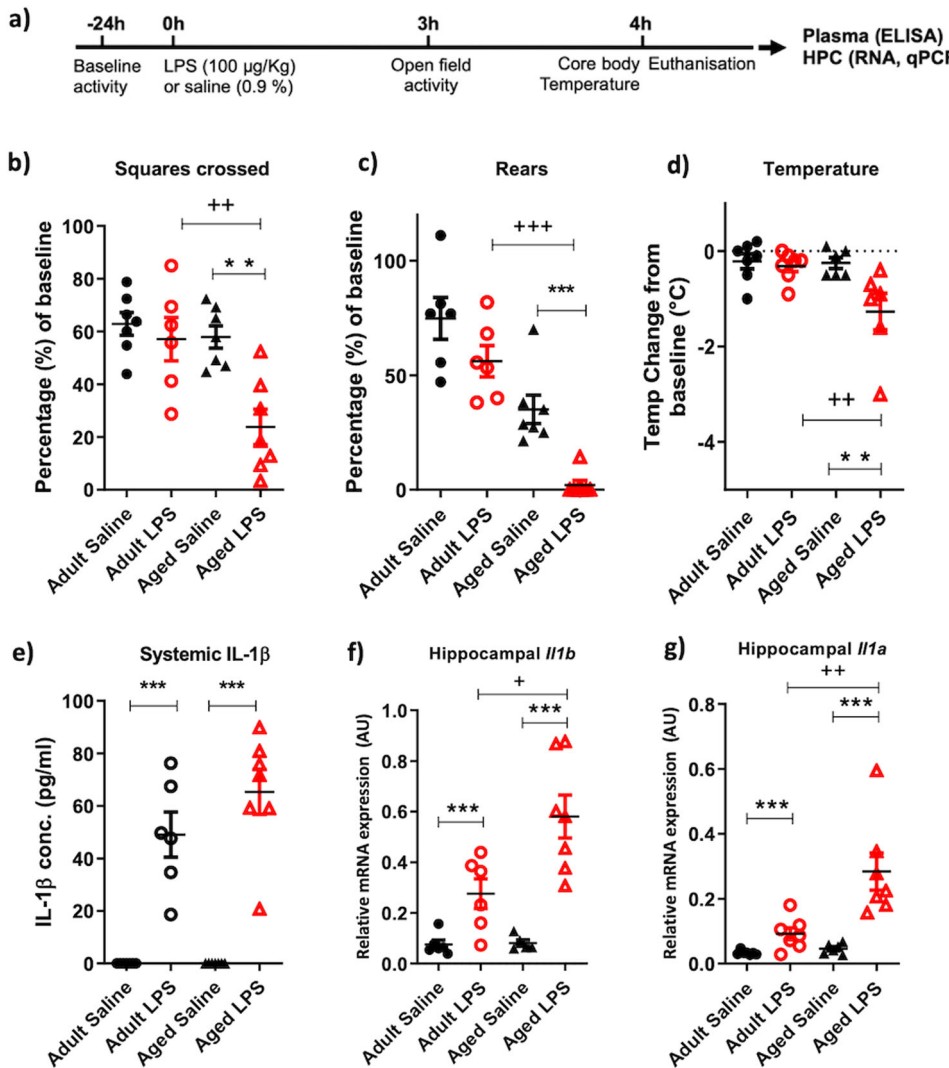

**Fig. 1 LPS induces exaggerated sickness responses and IL-1 responses in aged mice. a, b** Effect treatment with LPS (100 μg/kg i.p.) on sickness behaviour in adult (8 months) and aged (25 months) measured at 3-hours-post intraperitoneal LPS, measured by open field activity (**b**) distance covered (squares crossed) and (**c**) number of rears expressed as a percentage of baseline performance in adult and aged cohorts. **d** Effect of LPS (100 μg/kg) at 4-hours-post i.p. LPS treatment on core body temperature (°C). **e** Plasma IL-1β concentration at 4 hours post-LPS. **f, g** hippocampal *Il1b* and *Il1a* transcript expression at 4 hours post-LPS. All data are represented as mean ± SEM. All data were analysed by two-way ANOVA followed, upon significant main effects, by Bonferroni multiple comparison post-hoc tests ($n = 6,7$ independent animals), * denotes a statistically significant effect of treatment ($p < 0.05$),**($p < 0.01$), ***($p < 0.001$) by Bonferroni post-hoc test. + denotes significant difference between young and aged responses to LPS treatment ++($p < 0.01$), +++($p < 0.001$) by Bonferroni post-hoc test.

damage, as follows: *C1qa* and *C3* have been implicated in the engulfment of synaptic elements in several studies of multiple pathologies in mice. TREM2 and its adaptor protein DAP12 (encoded by *Tyrobp*) combine to facilitate recognition and engulfment of different damage motifs including apoptotic cells and lipids. *CD68* is a marker of the endosome/phagosome and is widely used as an indicator of phagocytic activity and *Clec7a* is a marker of the phenotype of primed/DAM microglia. All *p* values indicate Bonferroni post-hoc tests after significant main effects or interactions in 2-way ANOVA analysis. Both the cerebellum and hypothalamus demonstrated a clear induction of *Trem2, Tyrobp* and *Cd68* with age ($p < 0.001$), compared to adult controls (saline-injected). The cerebellum and hypothalamus also showed significant elevation of the complement components C1qα and C3 (Fig. 2a, b; $p < 0.001$; c,d: $p < 0.05$). The level of expression of all 5 of these genes was equivalent in saline and LPS-treated aged animals, indicating that this microglial activation arises as a

consequence of aging, independent of systemic inflammation, and is not further altered at 4 hours post-LPS.

The hippocampus displayed more subtle microglial activation with age across all of these markers. However, there was a significant interaction between age and LPS such that LPS did increase expression of these genes in aged mice but not adult mice. LPS induced significant increases in expression of *C1qa, C3, Trem2* ($p < 0.01$), *Tyrobp* ($p < 0.05$) and *Cd68* ($p < 0.01$), which were absent in LPS-treated younger animals.

The cortex showed more modest innate immune response activation. *Trem2* showed no significant alteration in expression with age or treatment while *Tyrobp* ($p < 0.01$) and *Cd68* ($p < 0.001$) were upregulated in aged animals treated with LPS. Thus, while age alone was not sufficient to induce changes, it did leave the animals susceptible to altered expression when challenged with a secondary stimulus. There was no significant alteration to the complement system genes *C1qα, C3* with age or LPS.

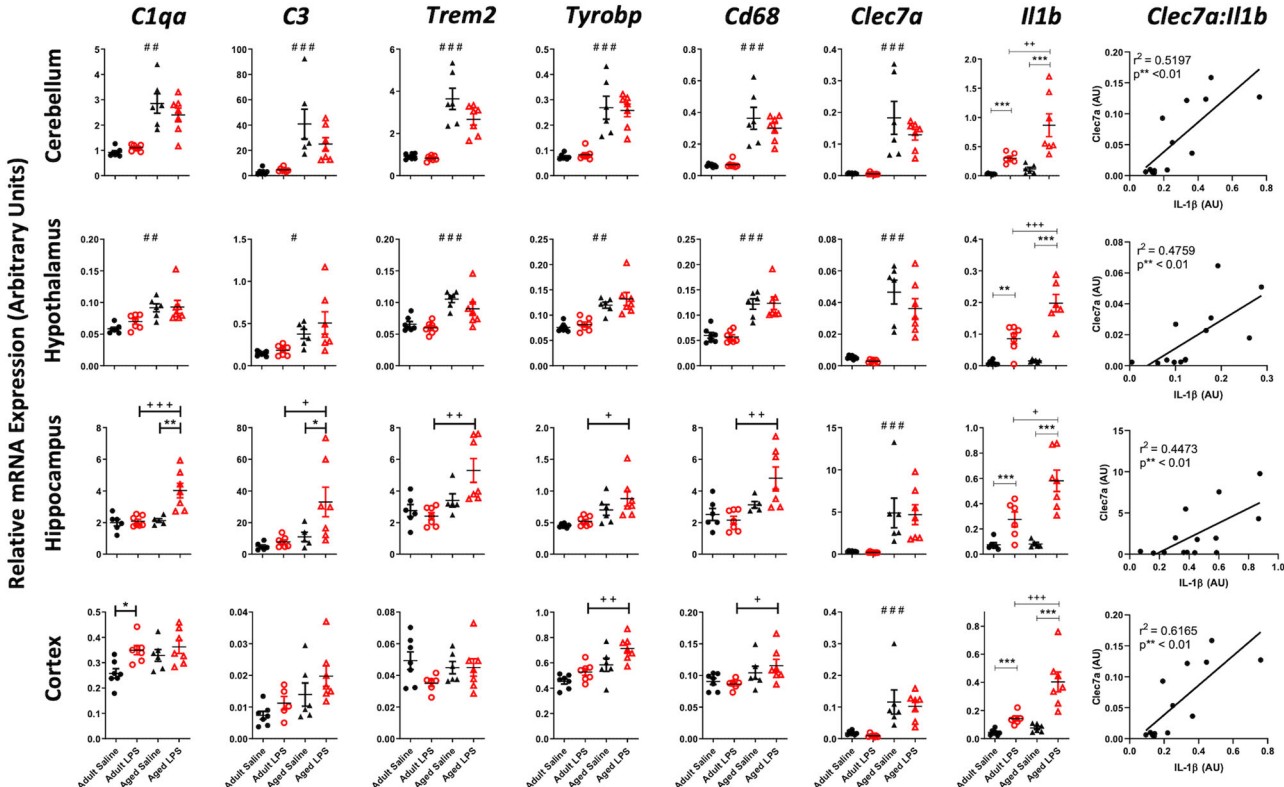

**Fig. 2 Microglial and complement system marker expression across the aged brain.** Effect of age (aged 8 months vs 25 months) on neuroinflammatory transcript expression (columns) in cerebellum, hypothalamus, hippocampus and prefrontal cortex (rows) under acute systemic challenge with LPS (100 μg/kg) i.p. treatment. All data are represented by mean ± SEM ($n = 6,7$ animals per treatment) and analysed by two-way ANOVA followed by a Bonferroni post-hoc test, # denotes a statistically significant main effect of age on transcript expression; # ($p < 0.05$), ## ($p < 0.01$), ### ($p < 0.001$). + denotes a statistically significant interaction between age and treatment (i.e. differential of effect of LPS in aged animals) + ($p < 0.05$), ++ ($p < 0.01$), +++ ($p < 0.001$).

Despite this regionally heterogeneous microglial signature the aged brain showed ubiquitous microglial priming, exemplified by the elevated expression of the priming 'signature' transcript[31] *Clec7a* ($p < 0.001$ in all regions) and exaggerated LPS-induced *Il1b*, the functional index of acutely stimulated, primed, microglia[32] ($p < 0.001$ in all regions). Moreover, levels of *Il1b* expression were strongly correlated with *Clec7a* transcripts in all 4 regions. Aged animals challenged with LPS also showed elevated expression of *Il1a* and *Tnfa* compared to adult challenged with LPS (supplementary Fig. 2).

**Age progressively increases risk for vulnerability to acute cognitive deficits.** We assessed the effect of age on susceptibility to acute cognitive dysfunction upon treatment with systemic inflammatory stimuli, LPS and poly I:C. Animals aged 5-7, 16-19 and 24-months of age (a separate cohort from those used in Fig. 1). Females weighed $21 + 2$, $24 ± 2$ and $24 ± 5$ g and males weighed $32 ± 2$, $29 ± 5$, $32 ± 3$ g respectively at these ascending ages) were trained on an 'escape from shallow water' version of the T-maze alternation task. This task has been specifically designed to test animals on working memory during concurrent sickness behaviour as previously described and validated by our group[24]. In short, since the task is aversively motivated and is scored dependent only on correct/incorrect choice of an escape arm, it is not confounded by low motivation, decreased exploratory drive or locomotor speed and is, therefore, a reliable measure of working memory function during acute sickness. Mice were trained for ≥10 blocks of 10 trials in order to reach the criterion baseline performance of ≥80% correct alternation.

All animals that were included in the study, regardless of age, were capable of learning the alternation strategy to escape the maze at baseline. Mice were then challenged with saline at day 0 followed by a 48 hour wash-out before intraperitoneal (i.p.) challenge with LPS (100 μg/kg), another 48 hour washout then challenged with the dsRNA viral mimetic poly I:C (12 mg/kg) before a final 48 hour washout before sacrificing the animal. Data were analysed by by mixed-model ANOVA. There was a main effect of age ($F_{2,49} = 7.64$, $p < 0.001$) across the time course of multiple treatments. Initial challenge with saline caused no significant impairment in performance in the maze (Fig. 3). Acute challenge with LPS significantly impaired cognition in 24-month-old mice compared to their younger 16-19 and 5–7-month counterparts at 3 and at 5 hours post-challenge (Bonferroni post-hoc, $p < 0.05$). A more robust effect was seen with the viral mimetic poly I:C at 2 ($p < 0.001$) and 4 hours ($p < 0.05$) post-challenge for the same 24 month old animals. However, the 16-19 month group also showed an impairment at 2 hours compared to 5-7 month old animals but by 4 hours post challenge had recovered to a level of performance equivalent to the youngest group. The data indicate that advancing age imposes progressive cognitive vulnerability to systemic challenge. Additionally there was an effect of sex on the response to LPS whereby, at 16-19 months of age, males were significantly more susceptible to acute disruption after LPS, but not after poly I:C. There was a main effect of sex ($F_{1,10} = 6.21$; $p = 0.031$) and a significant difference between males and females at 3 hours post-LPS ($p < 0.05$, Bonferroni post-hoc). These data are shown in supplementary fig. 3.

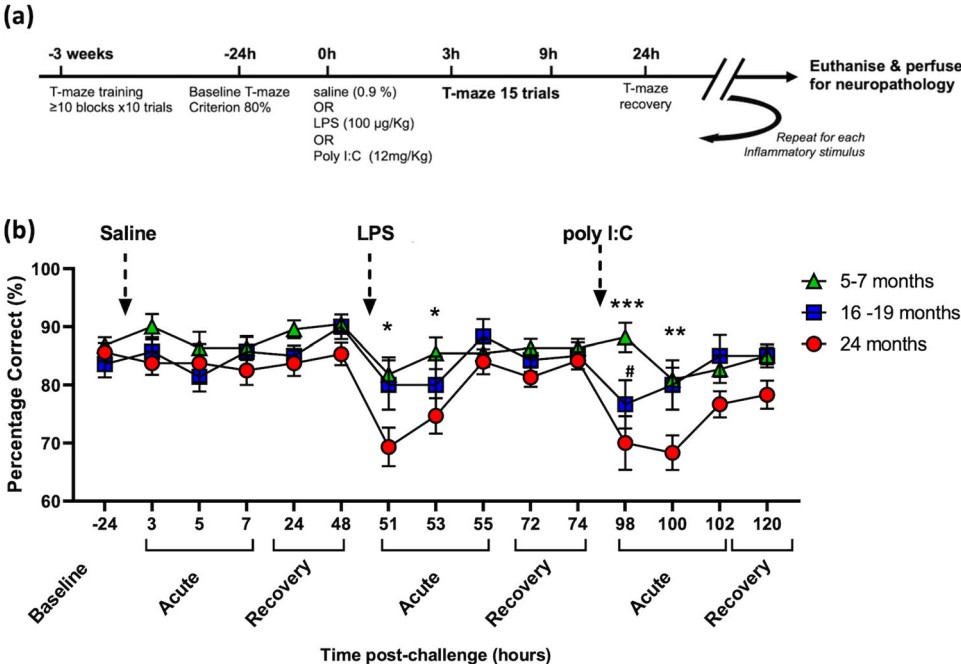

**Fig. 3 Age-dependent cognitive vulnerability to systemic inflammation-induced acute cognitive deficits. a** Schematic illustrating the experimental timeline. **b** Impact of consecutive saline, LPS (100 μg/kg i.p.) and Poly I:C (12 mg/kg i.p.) on working memory assessed by T-Maze. All data are represented graphically by mean ± SEM (Ages: 24 months $n = 13$, 16-19 months $n = 12$, 5-7 months $n = 22$), analysed by repeated measures two-way ANOVA, across the full time course with multiple comparisons by Bonferroni post-hoc test, * denotes treatment is significantly different between 24-month and 5-7-month cohorts *($p < 0.05$), ***($p < 0.001$), # denotes that treatment is significantly different between 16-19-month and 5-7-month cohorts #($p < 0.05$).

Given that there were some 24 month old animals that maintained good performance, and some young animals that showed impairments, it was apparent that age was not sufficient, alone, to explain the differential performance deficits in the 3 age groups and we therefore proposed that underlying changes in brain integrity, rather than age, may be the key determinant of cognitive performance under these acute stressors. Therefore, using these data we generated a cognitive frailty categorisation using cognitive scores from the acute response of all animals for which we had fixed tissue available to perform neuropathological analysis to the 2 acute inflammatory challenges (i.e. 3, 5 and 7 hours post-LPS and, 4- and 6 hours post-poly I:C). Their cognitive performance data were thus used to stratify the cohort into cognitively healthy "Resilient" ( < 7 errors) and cognitively vulnerable "Frail" ( ≥ 7 errors) animals for neuropathological analysis (Table 1). Greater than 7 errors indicated that an animal had scored <80% in one or more blocks of five trials they were assessed upon while under acute systemic inflammation and, indeed, all frail animals showed ≥2 blocks scoring 60% (see methods for further details). These classifications of old/young and cognitively frail/resilient were used to assess the relationship of cognitive frailty to neuropathological markers of microglial activation, white matter integrity and pre-synaptic terminal density.

**White matter microgliosis in cognitively resilient and cognitively frail animals**. To examine changes in microglial number and morphology, in young, cognitively frail and cognitively resilient brains, microglia from the hippocampus and cerebellum were labelled with the microglial markers IBA-1 and Pu1. Predictably, there was an age-dependent increase in microgliosis, but our analysis focussed on differences between aged animals when stratified by cognitive status. (Fig. 4). Changes in the microglia in the aged white matter regions, such as the fimbria and corpus callosum (CC), were much more

**Table 1 T-maze performance over 6 blocks of 5 trials, post-inflammatory stimulation.**

| | | LPS (100µg/kg) | | | Poly I:C (12mg/kg) | | | # of errors | cognitive vulnerability |
| | | Block 1 | Block 2 | Block 3 | Block 4 | Block 5 | Block 6 | | |
|---|---|---|---|---|---|---|---|---|---|
| Young (5-7 months) | Female | 80 | 80 | 80 | 100 | 80 | 80 | 5 | Resilient |
| | Female | 80 | 80 | 80 | 60 | 60 | 60 | 9 | Frail |
| | Female | 80 | 100 | 80 | 80 | 80 | 100 | 4 | Resilient |
| | Female | 100 | 80 | 100 | 100 | 100 | 100 | 1 | Resilient |
| | Male | 100 | 100 | 80 | 100 | 100 | 100 | 2 | Resilient |
| | Male | 100 | 80 | 80 | 80 | 80 | 80 | 5 | Resilient |
| | Male | 80 | 80 | 80 | 100 | 60 | 80 | 6 | Resilient |
| | Male | 80 | 100 | 100 | 100 | 80 | 80 | 3 | Resilient |
| | Male | 60 | 60 | 60 | 100 | 80 | 80 | 8 | Frail |
| | Female | 100 | 100 | 100 | 80 | 100 | 80 | 2 | Resilient |
| | Female | 80 | 80 | 80 | 80 | 80 | 80 | 6 | Resilient |
| | Female | 80 | 80 | 80 | 80 | 80 | 80 | 6 | Resilient |
| Aged (16-19 months) | Female | 100 | 100 | 100 | 100 | 100 | 100 | 1 | Resilient |
| | Female | 100 | 80 | 100 | 60 | 100 | 100 | 3 | Resilient |
| | Female | 100 | 100 | 100 | 80 | 80 | 100 | 2 | Resilient |
| | Female | 80 | 80 | 100 | 60 | 80 | 80 | 6 | Resilient |
| | Female | 80 | 100 | 80 | 80 | 80 | 100 | 4 | Resilient |
| | Female | 80 | 80 | 80 | 80 | 80 | 80 | 6 | Resilient |
| Aged (24 months) | Male | 60 | 60 | 100 | 60 | 60 | 60 | 9 | Frail |
| | Male | 100 | 80 | 80 | 60 | 60 | 80 | 7 | Frail |
| | Male | 80 | 80 | 80 | 80 | 100 | 100 | 4 | Resilient |
| | Male | 60 | 60 | 80 | 60 | 80 | 100 | 8 | Frail |
| | Male | 100 | 80 | 80 | 100 | 60 | 80 | 5 | Resilient |
| | Female | 60 | 80 | 80 | 60 | 80 | 80 | 8 | Frail |
| | Female | 80 | 80 | 80 | 60 | 80 | 100 | 6 | Resilient |
| | Female | 60 | 80 | 80 | 60 | 60 | 60 | 9 | Frail |
| | Female | 60 | 80 | 80 | 80 | 60 | 80 | 8 | Frail |
| | Female | 60 | 80 | 80 | 60 | 60 | 80 | 9 | Frail |

Pink indicates 2 errors in any given block of 5 trials (60%), white indicates 1 error (80%) and blue indicates no errors (100% performance) 7 errors or more defines that animal as cognitively frail.

striking than in grey matter areas, displaying shorter, thicker processes with large cell bodies consistent with previous reports of differential microglial activation between white and grey matter regions[20]. Aged cognitively frail animals showed more

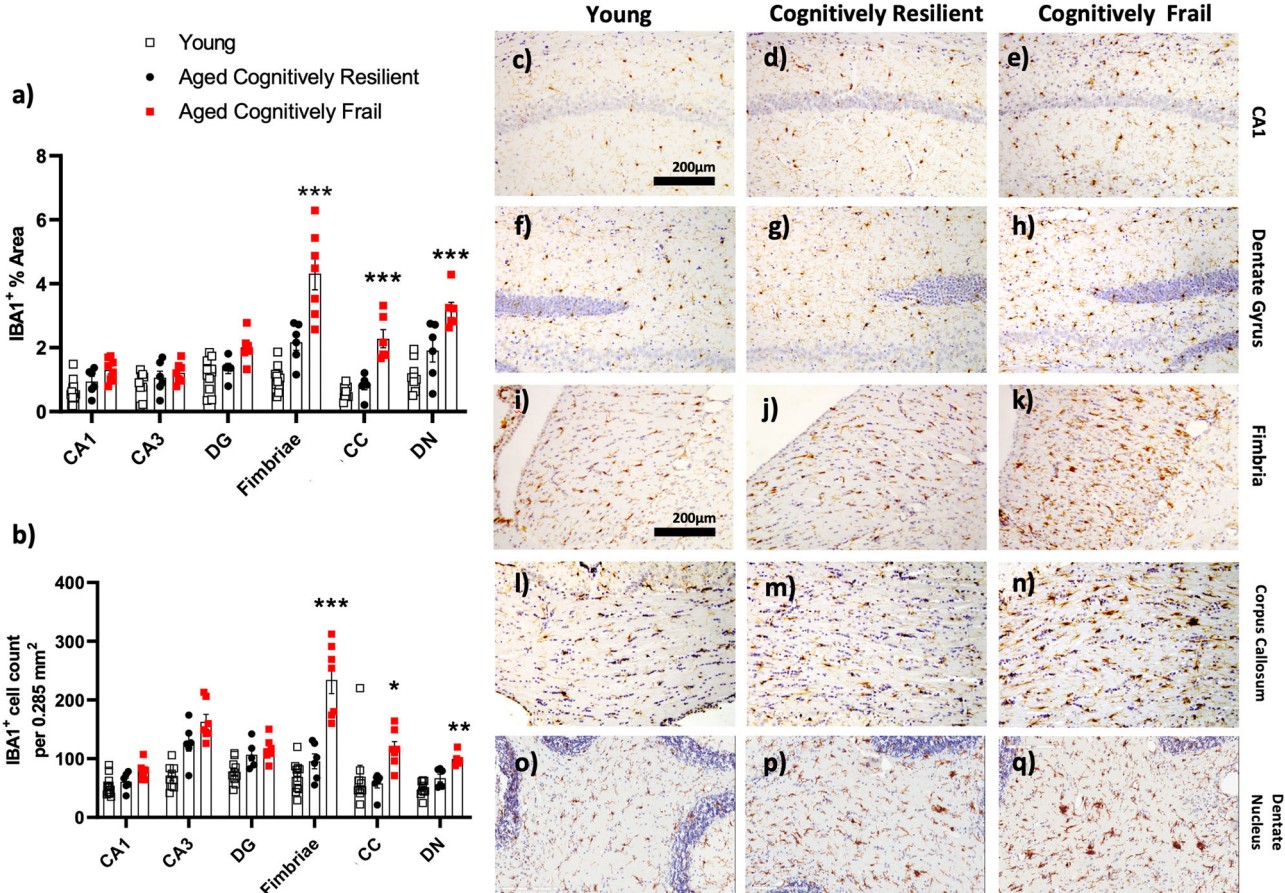

**Fig. 4 Relationship between cognitive status and histochemical and quantitative analysis of microglial activation.** Quantification of positively stained IBA-1+ microglia area (**a**) and count of IBA-1+ cells (**b**) in multiple areas of the hippocampal formation, categorised by age (young 5-7 months, aged includes all from 16-24 months) & cognitive frailty; quantified using Image J (NIH) at 20x. **c**–**q** Immunohistochemical analysis of microglial activation and morphology of microglia labelled with IBA-1 in the CA1 (**c, d, e**), dentate gyrus (**f, g, h**), fimbriae (**I, j, k**), corpus callosum (**l, m, n**), dentate nucleus (**o, p, q**). All data are represented by mean ± SEM and analysed by one-way ANOVA with Bonferroni post-hoc tests to detect microglial differences between young, (n ≥ 9), old cognitively resilient (n = 6) and old cognitively frail (n = 7) Statistically significant Bonferroni post-hoc differences between aged cognitively resilient and aged cognitively frail animals after significant main effects are denoted by *p < 0.05, **p < 0.01, ***p < 0.001.

substantial microglial changes compared to old cognitively resilient animals, showing increased cell number, larger cell soma, manifest as increased IBA1-positive area (represented quantitatively in Fig. 4a, b). These indices were only statistically significant in the white matter regions: fimbria, dentate nucleus (DN) of the cerebellum and at multiple levels on the anterior posterior (AP) axis of the corpus callosum (CC) (Fig. 4c-q). 'Early' and 'late' indicate more anterior (overlying striatum) and more posterior (overlying hippocampus) regions. All datasets were normally distributed and assessed by one-way ANOVA except Pu1 in the dentate nucleus, which was analysed using Kruskal Wallis. There were significant main effects of group with respect to both IBA-1-positive cell number and IBA-1 % stained area in the fimbria (F$_{2,21}$ = 37.66, 34.21 respectively), CC (F$_{2,18}$ = 4.49, 27.46 respectively) and DN (F$_{2,18}$ = 22.66, 18.29 respectively). Bonferroni post-hoc tests revealed significantly greater microgliosis in aged cognitively frail animals compared to cognitively resilient in the fimbria (counts: $p < 0.001$, area: <0.001), the CC (counts: $p < 0.05$ and area $p < 0.001$) and the DN (counts: $p < 0.01$, area<0.001). These findings were corroborated by a significant increase in the microglial specific nuclear marker Pu.1 in the same 3 regions (fimbria $p < 0.001 =$ F$_{2,14}$ = 17.43; CC $p < 0.001$, F$_{2,13}$ = 16.6.; DN $= p < 0.001$ nonparametric). Since not all animals were

available for Pu.1 analysis, this dataset is not directly comparable and is shown in supplementary fig. 4. There were no clear differences in microglial number/activation between young and cognitively resilient older animals.

**Decreased myelin integrity predicts cognitive frailty.** Given the increased microglial activation in white matter tracts we examined the relationship between white matter integrity and microgliosis. Brain sections were histologically stained with Luxol Fast Blue to quantify myelin density in the corpus callosum and fimbria (Fig. 5a-c and l-m). These were matched to IBA1 labelling performed in adjacent brain slices (Fig. 5d-f and o-q). Myelin density showed reductions in aged versus young mice, but this was statistically significant in CC but not the fimbria (Fig. 5g, r, $p < 0.01$ and $p = 0.48$ respectively). When comparing only by cognitive performance, irrespective of age, cognitively frail animals showed reduced white matter integrity compared to cognitively resilient animals in the CC and the fimbria (Fig. 5h, s) but this was significant only in the fimbria (CC $p = 0.06$; fimbria $p < 0.05$). Subsequently, stratifying the aged population by their cognitive status revealed a main effect of group for CC (Fig. 5i; $p < 0.05$, F$_{2,21}$ = 5.30). This was not the case for fimbria, despite low myelin integrity, since there was marked variability in the young and resilient populations (Fig. 5t; $p = 0.19$; F$_{2,19}$ = 1.79).

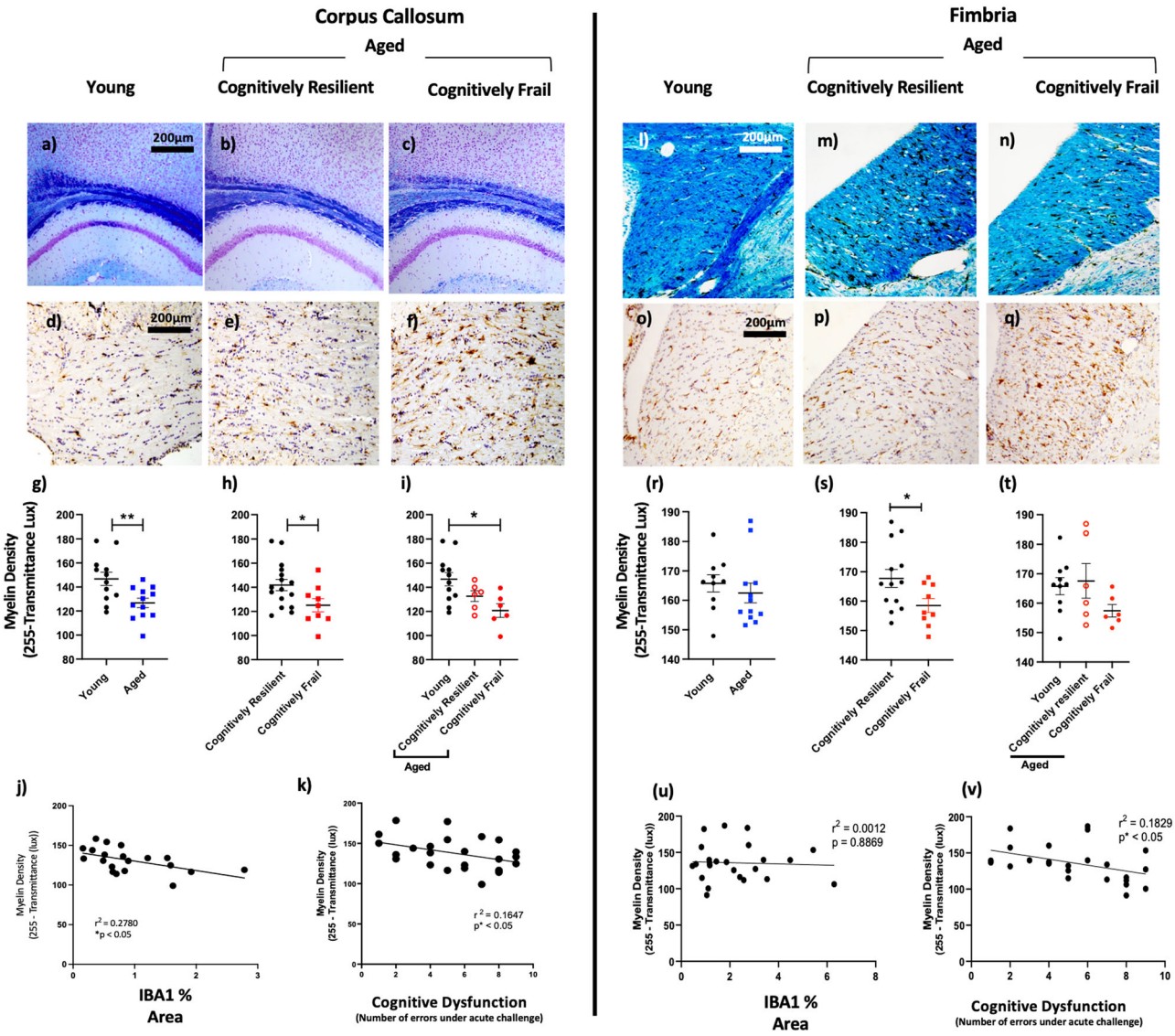

**Fig. 5 Integrity of white matter tracts of the hippocampal formation.** Photomicrographs of Luxol fast blue staining for myelin density in the hippocampus and corpus callosum (CC) and fimbria at 2.5X magnification. The effect of age ($n = 12$ young; 5-7 months, $n = 12$ aged/old includes all from 16-24 months) and cognitive frailty ($n = 16,8$) on myelin density and microgliosis were assessed by histochemical and quantitative analysis of LFB staining of the CC (**a–c**) and fimbria (**l–n**), and IBA-1+ immunostaining of CC (**d–f**) and fimbria (**o–q**). Quantitative analyses of myelin density as a function of different categorisations that distinguish the groups are presented in **g–i** for CC and in **r–t** for fimbria. All data are represented by mean ± SEM and analysed by student's t-test test (**g**, **h**, **r**, **s**) or one-way ANOVA with Bonferroni post-hoc tests (**i**, **t**) to detect effects of ageing and/or cognitive frailty with white matter measures (n = 10,6,6 for young, aged resilient and aged vulnerable respectively). Correlations assessed by Pearson's linear regression analysis of myelin density vs. % of IBA-1+ area in the CC and fimbria (**j**, **u**); myelin density vs. cognitive dysfunction (**k**, **v**). All images quantified using Image J (NIH) at 20x. Statistically significant differences between groups are indicated by $*p < 0.05$, $**p < 0.01$, $***p < 0.001$.

In the CC only aged cognitively frail mice were significantly different from young animals ($p < 0.05$).

Since these cognitively frail and resilient groups are categorical rather than continuous, we also conducted linear regression analysis on all age groups to assess the correlation between white matter integrity and acute cognitive dysfunction (# errors under acute systemic challenge). Pearson regression analysis demonstrated a negative correlation between cognitive dysfunction and white matter intensity in both the corpus callosum ($p < 0.05$, $r^2 = 0.1647$) and fimbria ($p < 0.01$, $r^2 = 0.1829$) (Fig. 5k, v). The corpus callosum also showed a significant correlation between reduced white matter integrity and % IBA1-positive area ($p < 0.05$, $r^2 = 0.278$ but this relationship was not present in the fimbria ($p = 0.8869$, $r^2 = 0.0012$) (Fig. 5j, u). Therefore, although

age increases white matter disruption and microgliosis in a general sense, the extent of this white matter pathology was predictive of the vulnerability of those mice to acute cognitive dysfunction upon exposure to secondary stressors.

**Cognitive frailty is weakly associated with presynaptic terminal density.** We also assessed for differences in pre-synaptic terminal density in aged and frail brains using labelling for the synaptic vesicle marker synaptophysin (Fig. 6a-c). Pre-synaptic terminal density was quite variable and categorisation by age failed to demonstrate any statistically significant reduction in synaptophysin density with age in this cohort (Fig. 6d). However, there was greater variability apparent in older animals such that

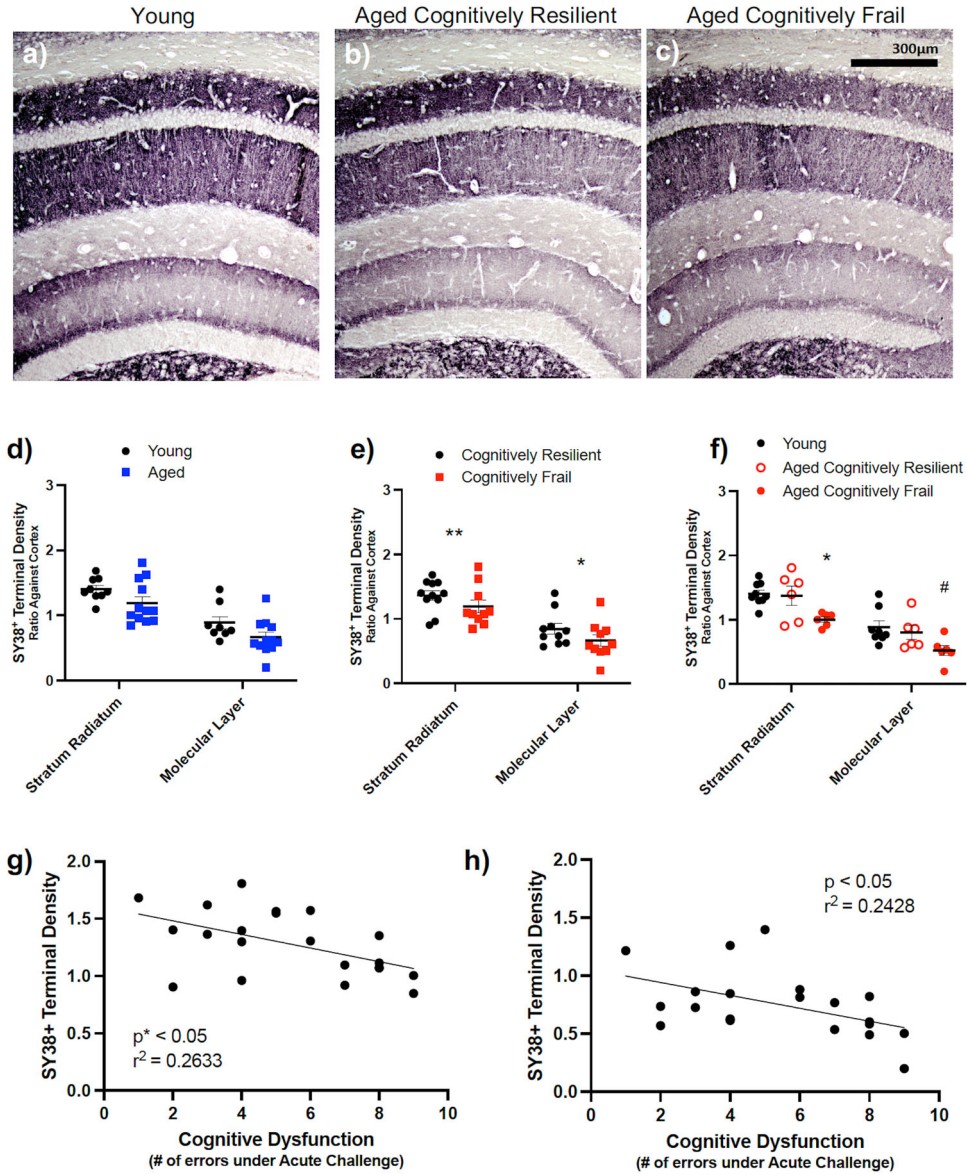

**Fig. 6 Quantitative and histochemical analysis of presynaptic neuronal integrity in the young and aged brain and with cognitive frailty. a–c** Presynaptic neuronal terminals density in the hippocampal layers stained with SY38 at 10x. **d–f** Sy38 density, quantified using Image J, plotted in Stratum Radiatum (SR) and Molecular layer (ML). All data represented by mean ± SEM and analysed by student's t-test test (**d**, **e**) to detect effects of age (n = 9,12; includes all from 16-24 months), cognitive frailty (n = 11,10). The statistically significant association of cognitive status with decreased density is indicated by *$p < 0.05$ (**e**). To assess association of aged cognitive frailty status (n = 9,6,6) with pre-synaptic terminal density, one-way ANOVA, followed by Bonferroni post-hoc tests were performed (**f**). Statistical significance is denoted by *$p < 0.05$ for reduced density in aged cognitively frail animals compared to aged cognitive resilient animals and #$p < 0.05$ for aged cognitively frail animals compared to young animals. Correlations of SY38[+] terminal density in the SR and ML vs. cognitive dysfunction assessed by Pearson's linear regression analysis (n = 21). All images quantified using Image J (NIH) at 20x, statistical significance indicated by *$p < 0.05$ (**g**, **h**).

chronological age was not a strong determinant of synaptic density. Stratifying by cognitive resilience/frailty across the cohort revealed significantly reduced pre-synaptic density in the stratum radiatum and molecular layers of the hippocampus in cognitively frail animals compared to the cognitively resilient (Fig. 6e) when assessed by student's t-test ($p < 0.01$ for radiatum, $p < 0.05$ for molecular layer and not significant for other layers). Moreover, when comparing young, old resilient and old cognitively frail by one-way ANOVA with Bonferroni post-hoc tests, there were significant effects of group in the radiatum (ANOVA $F_{2,18} = 3.57$, $p < 0.05$) and in the molecular layer ($F_{2,17} = 3.85$, $p < 0.05$). Aged, cognitively frail, animals were significantly lower than both aged cognitively resilient animals and young animals in the radiatum

($p < 0.05$), while these were only different to young animals in the molecular layer (Fig. 6f) in the stratum radiatum ($p < 0.05$). There were no other significant pairwise comparisons among the 3 groups.

Pearson regression analysis demonstrated a negative correlation between acute cognitive dysfunction and pre-synaptic terminal density in both the stratum radiatum ($p < 0.05$, $r^2 = 0.2633$) and the molecular layer of the dentate gyrus ($p < 0.05$, $r^2 = 0.2428$) (Fig. 6g, h).

## Discussion

Within the ageing population studied here, there is significant vulnerability to acute inflammation-induced cognitive

dysfunction that was heterogeneous within age categories. Stratifying animals as cognitively resilient or frail revealed that a greater degree of microgliosis in white matter regions conferred a significantly greater risk for cognitive dysfunction under acute inflammatory challenges. Despite clear brain region-dependent differences in neuroinflammatory signatures at baseline, all regions examined show microglia primed to show exaggerated brain cytokine responses to acute systemic inflammation. The vulnerability to acute cognitive dysfunction was significantly correlated with white matter microglial activation and with significantly reduced myelin integrity in those axonal tracts around the hippocampal formation. The cognitive frailty was also correlated with pre-synaptic terminal integrity in multiple layers of the hippocampus.

Consistent with the primed microglial signature in the CNS we observed exaggerated sickness behaviour in aged animals response to acute LPS treatment, despite plasma protein levels of IL-1β being comparable in young and old animals. This exaggerated sickness behaviour has been shown previously in the ME7 model of murine prion disease[33] and with age[22] when challenged with LPS and, based on thermoregulatory and brain injury responses[34], may constitute a maladaptive, IL-1β-dependent, response by the organism, divergent from the normal adaptive nature of sickness behaviour in resilient, healthy individuals[35]. Disproportionate effects of acute illness/injury in aged or vulnerable individuals may contribute to deleterious acute consequences including falls, delirium[36] as well as subsequent long-term cognitive decline and loss of independence[37–40]

Systemic inflammation, during acute illness, can produce acute cognitive dysfunction in older humans and this is best exemplified by delirium, which is a profound and rapid onset of fluctuating confusion and impaired awareness, sometimes referred to as acute confusion[36]. Age is a major risk factor for episodes of delirium upon acute illness and recent studies have shown that a patient's 'frailty' status significantly predicts their risk for experiencing delirium following surgery, injury or infection[8–14]. However, 'frailty' as conceptualised in the geriatric medical literature reflects the accumulation of multiple deficits within the body as a whole but it seems likely that, within a spectrum of age-associated deficits contributing to whole body frailty, brain vulnerabilities might contribute disproportionately to the risk for acute brain dysfunction such as delirium. Here we have demonstrated that while ageing conferred an increased vulnerability to acute cognitive dysfunction in the hippocampal-dependent T-maze task following systemic challenge this was not uniform and some animals displayed cognitive resilience to the acute stressor despite their advanced age. This model system then allows us to interrogate which features of brain integrity / degeneration may underpin the vulnerability to acute deficits. While we cannot state that the mice showing dynamic working memory deficits in the current study are actually experiencing delirium, the data illustrate the utility of studying normal aged mice as a model system to explore heterogeneous resilience and vulnerability to acute cognitive dysfunction, which is certainly useful to understanding the risk factors and pathophysiological routes to delirium.

Within the group of animals assessed for neuropathology, white matter microgliosis, loss of myelin integrity (as measured by Luxol fast blue staining) and loss of pre-synaptic terminals were significantly correlated with cognitive vulnerability. White matter integrity has previously been shown to decline with age and has been demonstrated to be associated with a decline in cognitive function and increased microgliosis in white matter-rich tissues with ageing and in many neurodegenerative disorders[21,41–44]. Meta-analyses of neuroimaging studies conclude that white matter disease and/or cortical atrophy are substantial risk factors for acute cognitive disruptions such as

delirium[27,45]. Moreover, a mouse model of delirium superimposed on evolving neurodegeneration showed that the prevalence, severity and duration of acute and fluctuating cognitive deficits was associated with the extent of underlying synaptic and axonal pathology[23]. Human epidemiological measures, in the same study, showed that human subjects' risk of delirium increased in a linear fashion as a function of baseline cognition. In diffusion tensor imaging studies of surgical patients at baseline (SAGES study), premorbid cognitive status, which predicted postoperative delirium, was associated with abnormalities in the corpus callosum, cingulum, and temporal lobe[28], while additional studies showed that impaired structural connectivity between posterior and frontal areas of the brain predicted post-surgical delirium[29]. The SAGES study also implicated abnormalities in the basal forebrain and we have previously shown that inducing loss of basal forebrain cholinergic neurons, in otherwise young healthy animals, is sufficient, alone, to increase risk for acute dysfunction similar to that shown here[25]. Given the importance of the myelin sheath in increasing velocity of action potentials[46] and synchronising signal transmission/integration within networks facilitating distributed functions[47] its correlation with increased susceptibility to acute stressor-induced cognitive dysfunction is highly suggestive of white matter integrity as a key predictor, and possibly determinant, of cognitive vulnerability.

Of interest in the current study, Luxol fast blue staining of myelin density in white matter tissue was inversely correlated with microgliosis in CC and cerebellar white matter tracts and that these features both correlated with increased vulnerability to acute cognitive dysfunction under acute inflammatory stress in the hippocampal-dependent T-maze task. Thus, there is a triad of correlated features indicating that microgliosis and loss of myelin may be responsible for a loss of resilience to acute stressors of the aged brain. The density of activated microglia in the cingulate gyrus and corpus callosum has been shown to be correlated with cognitive impairment in old monkeys[48]. Moreover several recent gene expression studies have highlighted the immune-vigilant phenotype of microglia in the white matter of the ageing brain[21,49] and demonstrated an enhanced microglial response following activation when compared to grey matter microglia[20]. Safaiyan and colleagues[21] describe TREM2-dependent, white matter-associated, microglia (WAM), present in activated nodules and engaged in phagocytosis/clearance of degenerating myelin[21,50] and there is evidence that these microglia may become senescent. These microglia share features with DAM[48], with primed microglia[31] and with the phenotype described here, characterised by expression of the core hub genes *Clec7a, Axl* and *Cd11c*. The current data add the significant finding that such age-dependent, microglia-associated, loss of myelin integrity foreshadows a loss of cognitive reserve and therefore may underpin the inability of some aged individuals (and not others) to demonstrate resilience in the face of an acute stressor. Although complex to quantify its contribution to age-associated cognitive dysfunction in given individuals, recent studies using Fingolimod or a minocycline and N-acetylcysteine combination treatment in ageing, multiple sclerosis and traumatic brain injury models have demonstrated simultaneous improvements in remyelination and cognition[51–54]. At this time we cannot say whether age-associated chronic microglial activation is inducing myelin disruption or whether age-related reduction of myelin density simply triggers a microglial response that does not contribute to demyelination and may even contribute to remyelination[55]. It is of significant interest to selectively disrupt microglial activation and function using antibodies or anti-sense oligonucleotides targeting microglial signalling to explore the relationship between myelin integrity, microgliosis and cognitive vulnerability. Likewise, other relationships within the white matter, including interactions

between oligodendrocytes and microglia may make important contributions that should be examined in future studies.

The regional differences in microglial activation observed here extend also to grey matter and also resonate with previous studies that showed more 'immune vigilant' microglia isolated from the cerebellum than in other regions in young mice. We found similar cerebellar and hypothalamic activation as evidenced by distinct age-dependent increases in basal expression of markers of microglial activation, *Trem2, Tyrobp*, and *Cd68*[19,56]. The hippocampus was described as losing its distinct transcriptional signature with advancing age[19,56] but we found no striking differences between adult and aged animals in these bulk RNA preparations. However, microglia in all regions examined appeared to be 'primed', as evidenced by elevated expression of the priming signature gene, *Clec7a*, and the associated strong upregulation of proinflammatory cytokines, *Il1b, Il1a* and *Tnfa*, in response to systemic LPS (Fig. 1, supplementary fig. 2). We propose that this is unlikely to be beneficial to the aged or vulnerable brain[18] and have previously shown deleterious consequences of this 'primed' CNS IL-1β response to systemic inflammation[34].

Here we showed significantly reduced Sy38+ pre-synaptic terminal density in several hippocampal layers including the stratum radiatum and the molecular layer in aged animals that were scored as cognitively frail in the hippocampal-dependent T-maze task compared to cognitively resilient age-matched animals. These results are consistent with previous findings that normal aging results in a region-specific vulnerability of synapses to loss and dysfunction[57] as well as an imbalance between excitatory and inhibitory synapses[58]. Early studies show strong regional and temporal overlap between loss of presynaptic terminals in the stratum radiatum of the hippocampus and microgliosis during neurodegeneration[59] and more recent studies show that microglia, via complement components C1q and C3, are actively involved in synaptic loss in models of AD[60]. Therefore it is plausible that the changes in grey matter microglial populations relate to age-dependent synaptic loss. However, correlations between synaptic density and cognitive function and between gross CA1 and dentate gyrus microgliosis and presynaptic terminal loss in dorsal hippocampal layers were relatively weak. Neither were there significant increases C3 or C1q in bulk RNA from the hippocampus. This may be because both synaptic loss and microgliosis were rather limited compared to that seen studies of neurodegenerative disease[59]. However, it may be of consequence for de novo synaptic disruption that C3 and C1q were markedly increased upon systemic LPS challenge and this was rather selective for the hippocampus (Fig. 2). The density of functional synapses is a crucial parameter in determining the efficacy of synaptic transmission and a reduced density of synaptic terminals is the strongest correlate of cognitive decline in AD[61]. Synaptic markers are also disrupted in CSF studies of patients with delirium in the context of infection[62] and synaptic density was strongly associated with vulnerability to acute cognitive dysfunction in an animal model of delirium[23]. With respect to inter-individual heterogeneity in cognitive resilience, it was also recently demonstrated that humans presenting with AD pathology but who remained cognitively resilient (i.e. asymptomatic for clinical dementia) exhibited greater dendritic spine density than those with dementia[63].

Finally, acute exacerbations of neuroinflammation can significantly exacerbate neuronal damage and degeneration in multiple models of neurodegenerative disease including MS, TBI, AD, and white matter injury[64–70] and can contribute to cognitive decline[39,40,71]. While the innate immune response might vary from region to region in the ageing brain, multiple regions may be equally at risk of harm from exaggerated production of pro-inflammatory cytokines, underpinned by the ubiquitously primed microglia described here (Fig. 2). Thus irrespective of the presence or absence of associations between microgliosis and reduced synaptic density in the current study, both of these features may be strong indicators of risk for acute dysfunction upon acute inflammatory stimulation.

It should be noted that due to the nature of frailty and resilience and their prevalence within the population studied here, aged cognitively resilient animals and young cognitively frail animals were not common in the study population and the statistical analysis is thus limited by this constraint. Moreover, although the overall study population was relatively well balanced for sex, most experiments did not have the statistical power to address whether there were sex-dependent effects and this needs further exploration. At 16-19 months old, females appeared more resilient to the cognitive-disrupting effects of LPS than males and, unfortunately, we did not have the neuropathological specimens to explore whether there were white matter and synaptic correlates of this differential susceptibility. Going forward it would be necessary to corroborate the relationships between microgliosis, white matter pathology and cognitive 'vulnerability' in larger populations, however the current data are consistent with previous reports from the literature as discussed above. Moreover, the current study is necessarily descriptive and we cannot, at this time, explore causation in the relationships described. It is intuitive that vulnerability to acute cognitive dysfunction is determined by underlying axonal and synaptic connectivity but describing specific molecular and structural determinants of this risk requires more detailed experiments. Ageing of animals with specific genetic manipulations targeting inflammatory pathways, or prolonged treatment with specific anti-inflammatory therapies will be necessary to unravel key determinants of cognitive vulnerability in the setting of ageing.

Taken together these data suggest that while ageing confers vulnerability to acute cognitive dysfunction upon immune activation, there is heterogeneous risk within the aged cohort and at least some of this may be explained by differential white matter microgliosis and loss of white matter integrity. Further work is required to determine the relative importance of reduced myelin and synaptic terminal density and their causal relationship with respect to microgliosis. Acute systemic inflammation, associated with delirium, is now known to significantly increase the risk of long-term cognitive decline and dementia[72,73], to exacerbate existing dementia[71,74] and to confer a greater risk of increased moribundity and mortality compared to patients of similar age admitted without delirium[72,75]. The recent COVID-19 global pandemic, with very high numbers of patients presenting with delirium or confusion, comorbid with fever[76], underlines the importance of understanding how systemic inflammation may contribute to acute and lasting cognitive disruption. It remains unclear whether episodes of delirium may have worse consequences for those with a low baseline brain integrity (more vulnerable to delirium) or those with more intact brains (wherein an episode of delirium may require a larger perturbation and therefore may produce a larger injury)[77]. With the estimated populations of aged individuals >65 yrs set to reach 1.5 billion by the year 2050[78,79] and with frailty a major modifier of the relationship between brain pathology and diagnosis of dementia[80] it is imperative that we interrogate acute cognitive dysfunction and its long-term consequences. As such, further study of cognitive resilience and frailty, and acute inflammatory mechanisms of acute dysfunction, will have utility in developing precision medicine approaches to treatment and prevention of delirium and associated long-term decline.

## Methods

**Animals**. Male & female C57BL/6 J mice (Harlan, Bicester, United Kingdom) were bred and housed in groups of five where possible at 21 °C with a standard 12:12 hour light dark cycle with access to food and water *ad libitum*. Separate cohorts of mice were used for sickness behaviour experiments (Figs. 1, 2) and cognitive/neuropathological experiments (Figs. 3–6). Mixed sex populations were used to provide a representative population although only cognitive experiments were powered sufficiently to examine sex differences. All animal experimentation was performed under a licence granted by Health Products Regulatory Authority (HPRA), Ireland, with approval from the local TCD Animal Research Ethics Committee and in compliance with the Cruelty to Animals Act, 1867 and the European Community Directive, 86/609/EEC, with every effort made to minimise stress to the animals. For sickness behaviour and molecular studies, mice were injected intraperitoneally (i.p.) with non-pyrogenic 0.9% sterile saline or 100 µg/kg of the bacterial endotoxin LPS (equine abortus, Sigma, L5886, Poole, UK) prepared in non-pyrogenic 0.9% sterile saline (Sigma, Poole, UK) and euthanised at 4 h (Fig. 1a). For cognitive experiments, animals were injected with sterile saline on day 0 (200 µl per 20 g body weight), with LPS (100 µg/kg) on day 3, followed by a 12 mg/kg dose of the viral mimetic Poly I:C (Amersham) prepared in non-pyrogenic 0.9% sterile saline (Sigma, Poole, UK) on day 5 (Fig. 3a). These doses mimic a mild to moderate infection rather than severe sepsis, eliciting small changes ( < 1 °C) in core body temperature in young healthy animals and acute working memory deficits only in mice with pre-existing pathology[81]. Core body temperature was measured using a thermocouple rectal probe (Thermalert TH5, Physitemp; Clifton, NJ, USA). In order to measure plasma glucose levels, animals were terminally anaesthetised with sodium pentobarbital (40 mg per mouse i.p. Euthatal, Merial Animal Health, Essex, UK), on day 7, their thoracic cavity was opened, and blood sampled directly from the right atrium of the heart using a glucometer (Accucheck, Roche UK), giving blood glucose levels in mmol/L.

**Open field activity**. The open field was a grey PVC arena, 50x30x18 cm, divided into $10 \times 10$ cm squares. The individual mice were placed in a corner square, facing the walls, and observed for 3 min. The total number of squares crossed and the number of rears were recorded. Rearing is defined by the mouse standing on their hind legs, with the fore paws not touching the floor and with their core engaged and head raised so as to survey their environment in the open field maze and is a measure of exploratory activity. Adult (8-month) and Aged (25-month) animals were assessed for baseline activity 24 hours before injection with 100 µg/kg of LPS or sterile saline and assessed once again 3-hours post i.p. challenge with the bacterial endotoxin or control.

**T-maze working memory**. To assess hippocampal-dependent working memory we used the paddling T-maze task as previously described[24]. This 'escape from shallow water' maze was custom-designed with the explicit aim of testing working memory in mice during acute sickness behaviour. Since all mice completed all trials and the scoring measures only correct/incorrect choices rather than speed or time to exit, the task is not confounded by suppression of general locomotor activity. Briefly, each mouse was placed in the start arm of the maze with 1 arm blocked such that they were forced to make a left (or right) turn, selected in a pseudorandomised sequence (equal numbers of left and right turns, no more than 2 consecutive runs to the same arm). On taking the turn the mouse could escape from the shallow water

and was held in a holding cage for 25 seconds (intra trial interval) during which time the guillotine door was removed and the exit tube was switched to the alternate arm. The mouse was then replaced in the start arm and could choose either arm. The mouse must alternate from its original arm to escape. Correct trials were recorded when the mouse alternated from its original turn and exited the maze. On choosing correctly mice escape from the maze and are returned to their home cage. On choosing incorrectly, the mice were allowed to self-correct to find the correct exit arm. Three age groups of mice (5-7 months, 16-19 months and 24-25 months), balanced for sex, were trained for ≥ 10 blocks of 10 trials (20 minute inter-trial interval) in order to reach the criterion performance of 80% correct alternation before challenging them sequentially with saline, LPS, and Poly I.C. as described above. All animals that were included in the study, regardless of age, were capable of learning the alternation strategy to escape the maze at baseline. All animals were tested for 15 trials after each acute treatment.

Subsequent to testing animals on cognitive function after acute challenges, animals were designated cognitively resilient or cognitively frail based on the number of errors they showed during the acute inflammatory challenges. Seven or more errors across these 6 blocks of 5 trials was designated 'frail' since 1 error per block of 5 (i.e. 80%) is typically considered normal performance, and indeed was our criterion performance for inclusion in the study. Anything greater than 7 errors is, therefore, indicative of working memory dysfunction and all animals that we designated cognitively 'frail' showed scores of 60% in at least 2 blocks of 5 trials post-inflammatory stimulation.

**Quantitative PCR**. The isolation of total RNA, synthesis of cDNA and analysis of transcription by quantitative PCR were performed as previously described[82]. Briefly, after transcardial perfusion with 0.9% heparinised saline, the hippocampus, hypothalamus, cerebellum and pre-frontal cortex were dissected out and snap frozen in liquid nitrogen and stored at −80 °C. Total RNA was extracted using Qiagen RNeasy Plus™ mini kits, with Qiashredders (Qiagen, Crawley, UK, #74134, #79654) according to manufacturer's instructions. Contaminating gDNA was removed using the Qiagen RNase-free DNase I enzyme (Qiagen #79254). RNA yields were determined by spectrophotometry at 260 and 280 nm and stored at −80 °C. Using a High Capacity cDNA Reverse Transcriptase Kit (Applied Biosystems, Warrington, UK), cDNA was synthesised using 200 ng of total RNA in a 10 µl reaction volume. 1 µl of the reverse transcription reaction was used for quantitative PCR. Reagents were supplied by Applied Biosystems (SYBR® Green PCR Master Mix; 4364344) and Roche (FastStart Universal Probe Master [Rox], Lewes, UK; 04914058001). Assays were designed using Primer Express software and published sequences for the genes of interest. All primer sequences are listed in Table 2. Assays were quantified using a relative standard curve, as previously described[82] constructed from cDNA, synthesised from 1 µg total RNA isolated from mice showing robust expression of all target transcripts of interest.

**ELISA for cytokines**. Under terminal anaesthesia, blood was collected directly from the right atrium into heparinised tubes, was centrifuged at 3000 rpm for 15 min at 4 °C and the remaining plasma aliquoted and stored at −20 °C. Samples were then analysed for IL-1β protein expression using a Quantikine kit (R&D systems, Minneapolis, MN, USA, MLB00C) as per manufacturer's instructions.

**Immuhistochemistry and histology**. A subgroup of the animals assessed for performance in the T-maze working memory task

**Table 2 qPCR primer & probe sequences.**

| Target | Accession Number | Oligonucleotide | Sequence | Amplicon Size (bp) |
|---|---|---|---|---|
| 18 S | NR_003278.3 | Forward | 5′-CGCCGCTAGAGGTGAAATTCT-3′ | 67 |
| | | Reverse | 5′-CATTCTTGGCAAATGTCTTTCG-3′ | |
| C1qα | NM_007572.2 | Forward | 5′-GCCGAGCACCCAACGGGAAGG-3′ | 268 |
| | | Reverse | 5′-GGCCGGGGCTGGTCCCTGATA-3′ | |
| C3 | NM_009778.2 | Forward | 5′-AAAGCCCAACACCAGCTACA-3′ | 115 |
| | | Reverse | 5′-GAATGCCCCAAGTTCTTCGC-3′ | |
| Cd11b | NM_001082960.1 | Forward | 5′-TCATTCGCTACGTAATTGGG-3′ | 71 |
| | | Reverse | 5′-GATGGTGTCGAGCTCTCTGC-3′ | |
| Cd68 | NM_009853 | Forward | 5′-CAAGGTCCAGGGAGGTTGTG-3′ | 75 |
| | | Reverse | 5′-CCAAAGGTAAGCTGTCCATAAGGA-3′ | |
| | | Probe | 5′-CGGTACCCATCCCCACCTGTCTCTCTC-3′ | |
| Clec7a | NM_020008.3 | Forward | 5′-CCCAACTCGTTTCAAGTCAG-3′ | 82 |
| | | Reverse | 5′-AGACCTCTGATCCATGAATCC-3′ | |
| Csf1r | NM_001037859.2 | Forward | 5′-GCAGTACCACCATCCACTTGTA-3′ | 140 |
| | | Reverse | 5′-GTGAGACACTGTCCTTCAGTGC-3′ | |
| Cxcl10 (IP-10) | M33266.1| | Forward | 5′-GCCGTCATTTTCTGCCTCAT -3′ | 127 |
| | | Reverse | 5′-GCTTCCCTATGGCCCTCATT-3′ | |
| | | Probe | 5′-TCTCGCAAGGACGGTCCGCTG-3′ | |
| Il1a | NM_010554 | Forward | 5′-TGTTGCTGAAGGAGTTGCCAG-3′ | 150 |
| | | Reverse | 5′-CCCGACTTTGTTCTTTGGTGG-3′ | |
| Il1b | M15131 | Forward | 5′-GCACACCCACCCTGCA-3′ | 69 |
| | | Reverse | 5′-ACCGCTTTTCCATCTTCTTCTT-3′ | |
| Irf7 | NM_016850 | Forward | 5′-CGAGGAACCCTATGCAGCAT-3′ | 108 |
| | | Reverse | 5′-TACATGATGGTCACATCCAGGAA-3′ | |
| Tnf-α | M11731 | Forward | 5′-CTCCAGGCGGTGCCTATG-3′ | 149 |
| | | Reverse | 5′-GGGCCATAGAACTGATGAGAGG-3′ | |
| | | Probe | 5′-TCAGCCTCTTCTCATTCCTGCTTGTGG-3′ | |
| Trem2 | NM_031254 | Forward | 5′-TGTGGTCAGAGGGCTGGACT-3′ | 68 |
| | | Reverse | 5′-CTCCGGGTCCAGTGAGGA-3′ | |
| | | Probe | 5′-CCAAGATGCTGGGCACCAACTTCAG-3′ | |
| Tyrobp | NM_011662.2 | Forward | 5′-CGTACAGGCCCAGAGTGAC-3′ | 91 |
| | | Reverse | 5′-CACCAAGTCACCCAGAACAA-3′ | |

were transcardially perfused with 0.9% heparinised saline followed by 10% neutral buffered formalin (Sigma, Poole, UK) for approximately 15 minutes/50 mls of fixative. Brains were then removed and post-fixed in the same fixative for 3 days before embedding in paraffin wax. Coronal sections (10 μm) of brains were cut on a Leica RM2235 Rotary Microtome (Leica Microsystems, Wetzlar, Germany) at the various positions along the anterior–posterior axis from Bregma, including medial septum, hippocampus and cerebellum, and floated onto electrostatically charged slides (Menzel-Glaser, Braunschweig, Germany) and dried at 37 °C overnight. All sections for immunohistochemistry were quenched for 20 minutes in 1% hydrogen peroxide in methanol. For IBA1 staining sections were microwaved in citrate buffer (pH 6) for 2 × 5 minutes and treated with 0.04% pepsin in 0.01 M HCl for 20 minutes prior to blocking in 10% normal rabbit serum. For Pu1 staining sections were microwaved in citrate buffer (pH 6) for 2 × 5 minutes and treated with 1% Trition for 20 mins prior to blocking in 10% goat serum. For synaptophysin, sections were treated with 0.2 M boric acid, pH 9, 65 °C for 30 min and cooled to room temperature before blocking in 10% normal horse serum. Primary antibodies were used in the following concentrations polyclonal goat anti-mouse IBA1 (ab5076) 1/2000, polyclonal rabbit anti-mouse Pu1 (CST226), monoclonal mouse anti-mouse SY38 (MAB5258) 1/2000, and incubated overnight at 4°C. Sections were then incubated in appropriate biotinylated secondary antibody at 1/100 (Vector Labs) and developed using the ABC method with hydrogen peroxide and diaminobenzidine (DAB) as the substrate. For synaptophysin staining, the DAB reaction was carried out in the presence of ammonium nickel chloride (0.06% w/v) to intensify staining. In the case of IBA1 a nuclear counterstain using Harris

Haemotoxylin counterstain was applied. For luxol fast blue staining sections were dewaxed and partially re-hydrated in 95% EtOH sections before incubating in a 0.05% luxol fast blue solution at 56 °C overnight. Slides were then differentiated in a 0.05% lithium carbonate solution followed by 70% EtOH until grey and white matter were clearly defined. Slides were then incubated for 5 minutes in the nucleic counterstain, 0.1% cresyl violet solution. For IBA1 analysis photomicrographs were taken at 20X under constant illumination. To eliminate background cell and objects, images were converted to 8-bit in Image J software (NIH, Bethesda, Maryland 20810, USA). and thresholds established to eliminate non-labelled background and preserve positively stained cells presence and morphology. A mask was drawn around the region of interest, excluding irrelevant regions or damaged areas of the section. Particles under a certain size (250 pixels) were eliminated, leaving only the microglia to analyse and obtain a count of and the total area of the mask which they occupied. All counts were then normalised to a uniform area so as to be comparable between sections. A similar approach was taken for analysis of the microglia specific marker Pu.1 as was conducted for IBA-1 above with the exception that particles under 50 pixels in size were eliminated due to the fact this nuclear stain is a far smaller area of staining than IBA-1. The density of synaptophysin staining was quantified by pixel density analysis on digitally captured images using ImageJ image analysis software (NIH, USA, http://imagej.nih.gov/ij/). To normalise values for variable labelling intensity, transmittance in the regions of interest (ROI) were subtracted from those for the corpus callosum and divided by transmittance in the overlying cortex to standardise for differences in density of labelling from section to section. Thus, the density of pre-synaptic terminal density labelling was quantified

according to the calculation: (transmittance corpus callosum – transmittance region of interest) divided by (transmittance corpus callosum – transmittance cortex). To assess the associations between white matter integrity changes and age or declining cognitive resilience, the light transmittance was quantified using the measure function and subtracted its value from that of pure white (255) to give a metric whereby high density (minimal light transmittance) gave a greater value than a region of low density (high light transmittance). Thus, white matter density staining was quantified according to the calculation: 255 – (Transmittance of ROI). To minimise the contribution of prior IBA1[+] labelling to reduced light transmittance in sections subsequently stained with Luxol fast blue, images were first processed in ImageJ by separating out the red and green spectra leaving only the blue light channel in the image for processing of Luxol Fast blue staining of myelin.

**Statistics and reproducibility**. Data were systematically assessed for homoscedasticity by Shapiro-wilk normality test and, in all cases where appropriate, parametric data were represented as mean ± SEM. For all sickness, temperature, ELISA and qPCR data, two-way ANOVA analyses were performed using age (young vs aged) and treatment (saline vs LPS 100 μg/kg) as between subjects factors. Bonferroni post-hoc tests were performed following significant main effects in ANOVA analysis. Pearson linear regression analysis was performed to determine whether *Clec7a* gene expression correlated with *Il1b* expression and significant *p*-values and $r^2$ values are reported. Longitudinal assessment of cognitive vulnerability to dysfunction over 48 hours pre and post i.p. challenge with saline, LPS (100 μg/kg) and Poly I:C (12 mg/kg) was analysed by a mixed-model ANOVA with age (5-7 months, 16-19 months, 24 months) as the between subjects' factor and time (hours pre/post challenge) as the within subjects' factor. This was favoured over standard repeated measures because 5 animals of 52 were withdrawn before poly I:C challenge due to weight loss. Bonferroni post-hoc tests were performed following significant main effects by ANOVA. Immunohistochemical cell counting and positively stained area were analysed by Student's t test for parametric data or Mann Whitney U test for non-parametric data. Factors used were age (young vs aged) and cognitive vulnerability (cognitively resilient vs cognitively frail). Pearson linear regression analysis was performed to determine whether myelin density (light transmitted) correlated with IBA1+ area or cognitive dysfunction, and whether synaptic density (light transmitted) correlated with cognitive dysfunction (assessed as number of errors under acute challenge), and significant p-values and $r^2$ values were reported.

### Data availability
Numerical source data for all graphs in the paper can be found in the supplementary data 1 file.

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

## Acknowledgements
This research was supported by an NIH R01 (AG050626) granted to CC. DH was supported by a College Award from the Trinity Foundation.

## Author contributions
D.H. performed behavioural, molecular and neuropathological experiments, analysed data and co-wrote the manuscript. C.M. performed cognitive experiments. C.M.cA., R.P., P.-L.H., J.L., L.T. and A.B.L.R. performed neuropathological analyses. C.C. conceived/designed the experiments, analysed data and co-wrote the manuscript.

## Competing interests
The authors declare no competing interests.
