## [Peer Review File · Communications Biology]

Reviewers' comments:

Reviewer #1 (Remarks to the Author):

This manuscript aims to expose underlying mechanisms (increased microgliosis and reduced white matter) to explain the heterogeneity in susceptibility to acute cognitive dysfunction in aged mice. It's an interesting hypothesis, but the experimental details and which/how the data are analyzed are hard to follow. Detailed comments follow.

1. Key methodological details are lacking. Under "Animals", the methods section states that mice received saline on day 0, LPS on day 3, Poly I:C on day 5 and euthanized on day 7, but it is unclear when, during this regimen, the mice were behaviorally tested. A schematic description of the timeline of experimentation would be helpful.
2. This lack of methodological detail comes into play for understanding the open field results. Saline-treated mice are being compared to LPS-treated mice, but presumably they were injected at different time points (saline at D0, LPS at D3). Was this behavior done before Poly I:C was injected? And how were squares crossed and rears calculated. The Y axis says Percentage of baseline, but it is not clear what is being used as baseline? If it is a percentage of their saline-injected behavior, why aren't the saline animals at an average of 100? Were mice placed in the open field once before their injections? This is confusing and needs to be clarified.
3. With regard to IL1 data in periphery vs hippocampus, it should be made clear that circulating levels were measured as IL1beta protein while hippocampal levels were measured as il1 gene transcripts. Thus, they cannot be compared. So in making conclusions, the authors should take care to keep this difference in mind.
4. The authors need to provide context for why they examined classical complement cascade, cd68, and Tyrobp gene transcription. This was not clear until nearly the last page of the manuscript.
5. An important detail of the present study is that data presented come from a single snapshot in time (3-4 hours post-LPS for sickness behaviors). There is evidence that sickness behaviors (temperature and activity) have different temporal profiles between young and aged rodents (<https://pubmed.ncbi.nlm.nih.gov/19486645/>) and depending on when one looks, results could look very different. Moreover, it could be that different regions of the brain could become reactive at different timepoints following peripheral activation. It would be worthwhile for the authors to acknowledge this point.
6. T-maze alternation working memory was conducted concurrently while the animals were sick and thus, it is impossible to tease apart cognitive function from sickness behavior-related factors such as fatigue, motivation, etc. thus any conclusions related to cognitive function should be softened. Also, how was a score of <7 errors deemed to represent "frailty"? What's the justification for this? Was it arbitrary or based on previous research?
7. For the white matter microgliosis and myelin data, why did the authors not include the 16-19 month-old group, stratify their cognitive vulnerability, and show their microgliosis & myelin data stratified by frailty group? It is also not clear why they didn't show the frail young adult brains. An n=3 of cognitively intact aged mice is not a fully powered experiment.
8. It would be great to see the data breakdown of males vs females? Unless a well-powered study has established that there are no sex differences in these groups they should not be lumped together, as this could mask effects from one of the sex groups.

Minor issues:

1. Under Quantitative PCR section, please indicate what mice were transcardially perfused with.
2. There are 2 Table 1s.
3. Age of mice in Results don't match the age of the mice listed in Table 1.

Reviewer #2 (Remarks to the Author):

In the manuscript "Heterogeneity in susceptibility to acute cognitive dysfunction in aged mice is underpinned by reduced white matter integrity and microgliosis" by Healy et al., authors investigated the sensitivity of aged brain to acute stress such as systemic inflammation and analysed the cognitive/motor skills and tissue pathological responses as a function of age progression. The region-dependent microglia responses, myelin, and synaptic densities assessed by qPCR/ IHC were correlated with cognitive parameters. Authors confirmed that aging is a risk factor for acute delirium and proposed that white matter integrity and synaptic connectivity are key factors underlining these age-dependent responses. The conclusions are supported by the obtained results and are generally consistent with previously reported data.

Minot comments:

- Numbering of pages/lines/figures would be helpful for a reviewer to read the text.

Methods:

- A scheme with overview of animal experiments would be helpful for readers to follow the experimental setup
- Describe how Iba1+ and Pu1+ cell counting was performed. The Pu1 antibody are not mentioned in the method section – they are first appeared in the Results
- Page 5: delete "Insert table here". In general, the polishing of the text is recommended

Results:

- Page 10: dividing corpus callosum on "early" and "late" to indicate anterior/posterior regions seems confusing especially in this manuscript, which is focused on age-dependent comparison.

Figures:

- Fig 1 D,E,F – some partially deleted text under the graphs titles is still visible – consider to improve the figure quality
- Figures 4 D,E and Fig 5 A-I and D-L are not mentioned in the text

Discussion:

- Page 13: Sentence "Likewise, we did not explicitly examine ..." needs revision

August 22nd 2023

To whom it may concern,

We would like to thank the referees for their critical review of the work. The manuscript has had substantial revisions in response to the points raised. We have:

- provided more detailed description of methods and more clarity on our discrete experimental cohorts
- performed new analyses of some of the results,
- included a whole new cohort to show sickness behaviour in young vs. aged animals at 6 and 24 hours post-LPS,
- increased the number of animals included the neuropathological analysis, with implications for the stratified analysis of microgliosis, white matter degradation and synaptic terminal density (so new data in figures 4, 5 and 6).
- provided a thorough explanation of the design of the T-maze and the basis for our confidence that it is robust to the confounds of concurrent sickness behaviour.

We have also made revisions to the discussion to make clear that we cannot state that our mice are experiencing delirium, but to also make clear why we believe the data are relevant to patients with delirium. Finally we have added to our limitations section of the discussion to highlight the shortcomings in fully addressing sex-dependent effects.

We respond to each comment, in a point-by-point fashion below:

Reviewers' comments:

Reviewer #1 (Remarks to the Author):

This manuscript aims to expose underlying mechanisms (increased microgliosis and reduced white matter) to explain the heterogeneity in susceptibility to acute cognitive dysfunction in aged mice. It's an interesting hypothesis, but the experimental details and which/how the data are analyzed are hard to follow. Detailed comments follow.

Comment 1. Key methodological details are lacking. Under "Animals", the methods section states that mice received saline on day 0, LPS on day 3, Poly I:C on day 5 and euthanized on day 7, but it is unclear when, during this regimen, the mice were behaviorally tested. A schematic description of the timeline of experimentation would be helpful.

Comment 2. This lack of methodological detail comes into play for understanding the open field results. Saline-treated mice are being compared to LPS-treated mice, but presumably they were injected at different time points (saline at D0, LPS at D3). Was this behavior done before Poly I:C was injected? And how were squares crossed and rears calculated. The Y axis says Percentage of baseline, but it is not clear what is being used as baseline? If it is a percentage of their saline-injected behavior, why aren't the saline animals at an average of

100? Were mice placed in the open field once before their injections? This is confusing and needs to be clarified.

Response (to comments 1 and 2)

Thank you for pointing out this significant omission. It is obvious in hindsight that we should have made this much more clear. The aged animals used for the sickness behaviour study, were euthanised early to perform the blood and brain PCR analysis. The cognitive behavioural analysis was performed on an entirely separate cohort, a significant number of which were then used for the neuropathology. We have included experimental design components to figures 1 and 3 to make this clear and have made it explicit in the text in both methods and results text (page 4 and page 10).

Baseline locomotor behaviour was measured at 24 hours before LPS/saline challenge in all animals in figure 1. Because most animals show some habituation upon repeat exposure to the open field arena, even animals challenged with saline show activity clearly less than 100% of their baseline score. This is now clarified on page 8.

Comment 3. With regard to Il1 data in periphery vs hippocampus, it should be made clear that circulating levels were measured as IL1beta protein while hippocampal levels were measured as il1 gene transcripts. Thus, they cannot be compared. So in making conclusions, the authors should take care to keep this difference in mind.

Response

This point is well taken. We make this distinction clear wherever we mention both (page 8).

Comment 4. The authors need to provide context for why they examined classical complement cascade, cd68, and Tyrobp gene transcription. This was not clear until nearly the last page of the manuscript.

Response

This is a fair point. We have made inappropriate assumptions about the readers' knowledge of currently prominent markers of microglial activation in the context of synaptic loss, phagocytosis and DAM phenotypes. C1qa and C3 are directly implicated in the engulfment of synaptic elements in several studies of multiple pathologies in mice. Trem2-encoded TREM2 and its adaptor protein DAP12 (encoded by Tyrobp) combine to facilitate recognition and engulfment of a number of different damage motifs including apoptotic cells and lipids. CD68 is a marker of the endosome/phagosome and is widely used as an indicator of phagocytic activity. Clec7a encodes Dectin1, which is part of the disease associated microglia phenotype and has, in our hands, been strongly correlated with the propensity of primed microglia to produce exaggerated Il1b/IL-1 β responses to secondary stimulation with LPS. We now briefly justify these transcript choices at the beginning of the results text that accompanies figure 2 (page 9).

Comment 5. An important detail of the present study is that data presented come from a single snapshot in time (3-4 hours post-LPS for sickness behaviors). There is evidence that sickness behaviors (temperature and activity) have different temporal profiles between young and aged rodents (<https://pubmed.ncbi.nlm.nih.gov/19486645/>) and depending on when one looks, results could look very different. Moreover, it could be that different regions

of the brain could become reactive at different timepoints following peripheral activation. It would be worthwhile for the authors to acknowledge this point.

Response

This is indeed a good point and one that we have now addressed using a third aged cohort. We now provide data to show that differences between adult and aged animals' responses to LPS largely persist over 24 hours. We have stated this explicitly in the results text on page 8 but since these animals are not the same animal cohort as those in the current figure 1, to avoid confusion we have included them in a new figure 1 in the supplementary data.

Comment 6. T-maze alternation working memory was conducted concurrently while the animals were sick and thus, it is impossible to tease apart cognitive function from sickness behavior-related factors such as fatigue, motivation, etc. thus any conclusions related to cognitive function should be softened. Also, how was a score of <7 errors deemed to represent "frailty"? What's the justification for this? Was it arbitrary or based on previous research?

Response

This is an extremely important point that is relevant to all studies in which cognition is assessed during acute sickness. Since several parameters including appetite, locomotor speed and tendency to explore are all significantly suppressed and since these are key elements that define performance in many cognitive tasks it makes most tasks unsuitable for use during sickness behaviour. Indeed we have, many years ago, written a review entirely focussed on this serious confound in cognitive studies during acute illness (Cunningham and Sanderson, BBI, 2008). Therefore we have been absolutely robust on this point as it has been a cornerstone of our work for many years:

We designed a custom-built 'escape from shallow water' version of the T-maze alternation task, designed specifically to test animals on working memory during concurrent sickness behaviour. This was previously described and validated by our group (Murray C, et al. Systemic inflammation induces acute working memory deficits in the primed brain: Relevance for delirium. Neurobiol Aging, doi:10.1016/j.neurobiolaging.2010.04.002). Because the maze is filled with shallow water the mice are motivated to solve and exit the maze even when quite sick. It is not a continuous alternation paradigm that relies on motivation to explore and, therefore, motivation is not a concern: the animal has a sample run and then a test run in which it must show alternation to escape (a correct alternation). Then this is repeated many times, in discrete trials over the 6 hours post-challenge, with the arm of exit in the sample run randomised so that the 'question' remains different every time it is run. In the case of the LPS and poly I:C experiments run here, the mice were tested 15 times in the hours after inflammatory treatment and have, therefore, returned 15 correct/incorrect responses. There are no incomplete trials: animals always complete the task.

Because the only measure reported is correct alternation, not speed or time in the maze, the cognitive performance of the mouse is measured on whether it makes an alternation (correct) or 'matches to sample' (incorrect). Nine correct trials out of 10 is captured as 90% alternation. Mice do not sit still, do not stop moving through the maze nor fail to engage with the task and in 15 years we have not observed any mice not exiting the maze when finding the exit. Therefore fatigue or motivation do not confound this shallow water T-maze. Thus we have intentionally and demonstrably teased apart fatigue and motivation from cognitive

impairment. We now make our attention to this potential confound more explicit (page 10 and briefly in the methods on page 5).

We hope this gives sufficient reassurance to the reviewer but, since this is a cornerstone for the validity of the whole study, we would be happy to discuss this further with reviewer and editor if necessary.

With respect to the 'cut off' for frail/resilient, 7 may seem arbitrary but it does have a rational basis. Across many years of research, normal mice perform at between 80 and 100% alternation so we do not regard 80% as cognitive impairment even though we do count those 2 errors out of 10 trials. Therefore in any group of normal performing animals we would expect several animals to have blocks of trials in which they score 80%. Thus across the 6 blocks of 5 trials that follow LPS + poly I:C, animals that make 1 error in each of those blocks (ie 6 errors in total) still remain in a range that would be regarded as normal (thus we accept <7 errors as remaining in the normal range). Any mouse showing 7 or more errors across those 6 blocks is now falling into a range where they have 2 errors or more in at least one of their 6 blocks post-inflammation. On that basis ≥ 7 errors are regarded as showing some cognitive frailty. This is now explained on page 5 and, in the results, on page 11.

Comment 7. For the white matter microgliosis and myelin data, why did the authors not include the 16-19 month-old group, stratify their cognitive vulnerability, and show their microgliosis & myelin data stratified by frailty group? It is also not clear why they didn't show the frail young adult brains. An $n=3$ of cognitively intact aged mice is not a fully powered experiment.

Response

It is reasonable to ask why the 16-19 month-old group were not initially included. Unfortunately the answer is a tale of misfortune. Most of the cohort, upon cutting, were placed in a 37 degree oven for drying after sectioning and collection on glass slides. The drying oven dangerously overheated and, honestly, burnt the tissue beyond use. The equipment has now been disposed of. There were some 16-19 month-old animals from whom intact tissue was available and we have now included all of these in the neuropathological study. These are now included in table 2 and in figures 4, 5 and 6. These animals were all in the resilient category and we have therefore been able to supplement the aged+resilient group, which previously had just $n=3$ and was rightly described as 'not fully powered'.

Therefore the figures 4, 5 and 6 have all been replotted and the statistics reanalysed, with consequent changes to the results text.

With respect to the frail young adult brain, we take the view that having just 2 young frail animals is insufficient to constitute a group for any meaningful analysis as a discrete group. Given their prior comments about our initial group of resilient older animals ($n=3$), we would think that this view would be shared by the referee. Based on many years of performing T-maze in young animals, after proper training we find very few young animals showing significant deficits on this task so young/frail is not a group that we either expected or observed. Nonetheless, these young 'frail' animals are included in the correlations in figures 5 and 6 so their contribution to overall effects is captured here.

Comment 8. It would be great to see the data breakdown of males vs females? Unless a

well-powered study has established that there are no sex differences in these groups they should not be lumped together, as this could mask effects from one of the sex groups.

Response

We are sympathetic to this viewpoint and our laboratory now routinely powers experiments for sex differences. However this was not the case at the time the current study was conducted and, therefore, the only experiment that has sufficient numbers to examine sex differences is the cognitive experiment. We have now analysed this experiment with sex as a factor. We find that the only sex difference emerges at 16-19 months, wherein the male animals are more susceptible to the acute challenge with LPS. This has now been stated clearly in the results text and a full comparison of males versus females across the full time course of treatment, at each age, is now shown in supplementary figure 3.

Minor issues:

1. Under Quantitative PCR section, please indicate what mice were transcardially perfused with.

Response. *Mice were perfused with 0.9% heparinised saline*

2. There are 2 Table 1s.

Response. *This has now been corrected, thank you.*

3. Age of mice in Results don't match the age of the mice listed in Table 1.

Response. *As discussed earlier the mice in figures 1 and 2 come from a different cohort to those in figures 3-6. We hope that the methods section and the figure legends for each experiment now capture these details clearly.*

Reviewer #2 (Remarks to the Author):

In the manuscript "Heterogeneity in susceptibility to acute cognitive dysfunction in aged mice is underpinned by reduced white matter integrity and microgliosis" by Healy et al., authors investigated the sensitivity of aged brain to acute stress such as systemic inflammation and analysed the cognitive/motor skills and tissue pathological responses as a function of age progression. The region-dependent microglia responses, myelin, and synaptic densities assessed by qPCR/ IHC were correlated with cognitive parameters. Authors confirmed that aging is a risk factor for acute delirium and proposed that white matter integrity and synaptic connectivity are key factors underlining these age-dependent responses. The conclusions are supported by the obtained results and are generally consistent with previously reported data.

Minor comments:

- Numbering of pages/lines/figures would be helpful for a reviewer to read the text.

Response: *Sorry for this omission, now fixed.*

Methods:

- A scheme with overview of animal experiments would be helpful for readers to follow the experimental setup

Response: *Apologies for this significant omission, we have now included experimental schematics in the 2 relevant figures (1 and 3).*

- Describe how Iba1+ and Pu1+ cell counting was performed. The Pu1 antibody are not mentioned in the method section – they are first appeared in the Results

Response:

Photomicrographs were taken at 20X under constant illumination. To eliminate background cell and objects, images were converted to 8-bit in Image J software (NIH, Bethesda, Maryland 20810, USA) and thresholds established to eliminate non-labelled background and preserve positively stained cells presence and morphology. A mask was drawn around the region of interest, excluding irrelevant regions or damaged areas of the section. Particles under a certain size (250 pixels) were eliminated, leaving only microglia included in the counted area/numbers. All counts were then normalised to an area of uniform transmittance in the same section in order to normalise for differences in overall intensity among different sections. A similar approach was taken for analysis of the microglia specific marker Pu.1 as was conducted for IBA-1 above with the exception that particles under 50 pixels in size were eliminated since Pu.1 labels only microglial nuclei and thus occupies a smaller area.

This text has been included in the methods section of the revised manuscript

- Page 5: delete “Insert table here”. In general, the polishing of the text is recommended.

Response: *Noted and fixed, thank you.*

Results:

- Page 10: dividing corpus callosum on “early” and “late” to indicate anterior/posterior regions seems confusing especially in this manuscript, which is focused on age-dependent comparison.

Response

The comparison among ages was the major motivation of the study but our assessment of microglial activation came early in the work, we screened the brain for striking areas of microglial activation. Since white matter was the most striking, compared to grey matter areas, we examined this in a number of regions. We think the clear effects in the 2 AP regions of the corpus callosum justifies including it in the analyses we share in the manuscript.

Perhaps the comment relates to the use of terms early and late, which may spark a ‘temporal’ association in readers? The ‘early’ corpus callosum was overlying the striatum while the ‘late’ corpus callosum was overlaying the hippocampus. In the results text we have now stated that the corpus callosum is “described here as ‘early’ and ‘late’ to indicate more

anterior and more posterior regions respectively". We have also removed the label LATE from the CC annotations on figure 4.

Figures:

- Fig 1 D,E,F – some partially deleted text under the graphs titles is still visible – consider to improve the figure quality

Response: *Thanks for pointing this out. We have been careful to tidy these.*

- Figures 4 D,E and Fig 5 A-I and D-L are not mentioned in the text.

Response: *Thank you for pointing this out. There has been a slight rearrangement of these 2 figures since we have added some new data. In re-creating these we have taken care to ensure that each panel is correctly labelled in both figure and figure legend and the labels follow as logically as possible. Some panels are referred to, in the text, only within a larger group but we still regard that it is helpful, for completeness to have a label for each individual picture since this may be helpful at the reviewing or discussion stage.*

Discussion:

- Page 13: Sentence "Likewise, we did not explicitly examine ..." needs revision

Response: *We have revised this sentence to articulate that the role of white matter oligodendrocytes merits examination.*

We hope we have answer all queries/comments.

Sincerely

REVIEWERS' COMMENTS:

Reviewer #1 (Remarks to the Author):

The authors should be commended for addressing each concern raised by the reviewers fully and respectfully. The revisions made have elevated the quality of the paper and I believe will be of great interest to those in the aging and behavioral neuroscience fields. Well done!

Reviewer #2 (Remarks to the Author):

The manuscript is significantly improved and all raised questions/comments were addressed in the revised version. Recommend to accept for publication.

A minor comment is mainly related to the text polishing, such as page 12, line 7, where the sentence seems not finished: "...and fimbria (Figure 5T; $p < 0.01$ $F_{2,18} = 2.26$) and."